

# Thermal and dissipative effects on the heating transition in a driven critical system

Kenny Choo[1], Bastien Lapierre[1*], Clemens Kuhlenkamp[2,3,4],
Apoorv Tiwari[1,5,6], Titus Neupert[1] and Ramasubramanian Chitra[7]

**1** Department of Physics, University of Zürich,
Winterthurerstrasse 190, 8057 Zürich, Switzerland
**2** Institute for Quantum Electronics, ETH Zürich, CH-8093 Zürich, Switzerland
**3** Department of Physics and Institute for Advanced Study,
Technical University of Munich, 85748 Garching, Germany
**4** München Center for Quantum Science and Technology,
Schellingstrasse 4, 80799 Munich, Germany
**5** Condensed Matter Theory Group, Paul Scherrer Institute,
CH-5232 Villigen PSI, Switzerland
**6** Department of Physics, KTH Royal Institute of Technology,
Stockholm, 106 91 Sweden
**7** Institute for Theoretical Physics, ETH Zürich,
Wolfgang-Pauli-Str. 27, 8093 Zürich, Switzerland

★ bastien.lapierre@uzh.ch

## Abstract

We study the dissipative dynamics of a periodically driven inhomogeneous critical lattice model in one dimension. The closed system dynamics starting from pure initial states is well-described by a driven Conformal Field Theory (CFT), which predicts the existence of both heating and non-heating phases in such systems. Heating is inhomogeneous and is manifested via the emergence of black-hole like horizons in the system. The robustness of this CFT phenomenology when considering thermal initial states and open systems remains elusive. First, we present analytical results for the Floquet CFT time evolution for thermal initial states. Moreover, using exact calculations of the time evolution of the lattice density matrix, we demonstrate that for short and intermediate times, the closed system phase diagram comprising heating and non-heating phases, persists for thermal initial states on the lattice. Secondly, in the fully open system with boundary dissipators, we show that the nontrivial spatial structure of the heating phase survives particle-conserving and non-conserving dissipations through clear signatures in mutual information and energy density evolution.

# 1 Introduction

Recent years have seen much progress in the understanding of out of equilibrium properties of many-body quantum systems. Two broad categories which have been explored extensively are (i) the dissipative dynamics of open-systems [1] and (ii) the dynamics of periodically driven systems [2–4]. Dissipation, in general, engenders an irreversible non-unitary evolution of the quantum system towards a steady state. The interplay between unitary Hamiltonian evolution and dissipation can lead to dissipative phase transitions via nonanalyticities in the steady state [5]. Understanding the role of dissipation is especially relevant for quantum simulation platforms which permit the realization of a multitude of theoretical phenomena and out of equilibrium phases not accessible in standard solid state systems [6–8]. Tailored dissipation can also be used as a resource for quantum state engineering of many-body phases [9–18].

In parallel, there has been enormous progress in harnessing the potential of periodic driving to generate new classes of out of equilibrium phenomena [4,19,20]. Well-known examples include discrete time crystals [21–23], anomalous Floquet-Anderson insulators [24], synthetic dimensions [25], synthetic gauge fields [26] to name a few. However, a fundamental issue underpins the potential success of such quantum engineering endeavours. In the absence of a fundamental energy conservation law in a driven system, an important question concerns whether the system can in principle absorb energy from the periodic drive and heat up. Past work seemed to indicate that non-integrable and interacting systems tend to heat up to a featureless infinite temperature state [27], when subject to a drive, whereas integrable systems, by virtue of their many conserved quantities do not heat up but tend to an asymptotic state governed by a periodic Gibbs ensemble [28, 29]. Exceptions to this paradigm include some integrable disordered systems [30] as well as many-body localized systems [31] which evade heating. The dynamics of heating has been recently explored experimentally in Bose-Einstein condensates (BECs) in driven optical lattices [32].

Recently, this interplay between integrability and interactions and its relevance for non-equilibrium dynamics was explored in a generic class of critical quantum systems in one spatial dimension. The underlying quantum critical system is described by a conformal field theory (CFT) in the long wavelength limit. A particularly interesting class of driving protocols in-

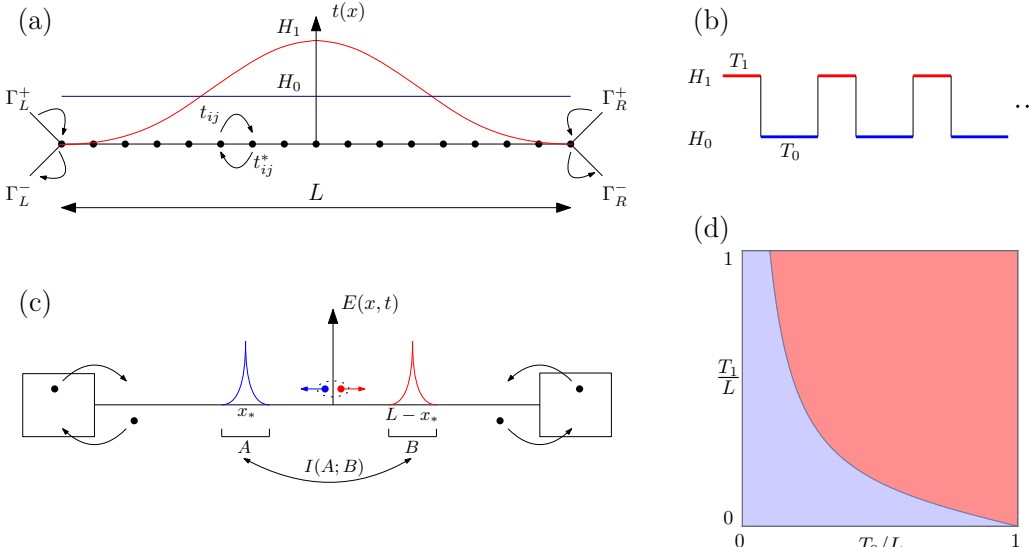

Figure 1: (a) Two Hamiltonians used to construct the Floquet drive: homogeneous Hamiltonian $H_0$ and sine-square deformed Hamiltonian $H_1$ on a free fermion chain of length $L$, with dissipators placed at the boundaries of the chain and characterised by dissipation rates $\Gamma_{L/R}^{\pm}$. (b) Illustration of the two step drive $H(t)$ between $H_0$ and $H_1$. (c) Sketch of the emergent spatial structure in the energy density $E(x,t)$ in the heating phase, with two hotspots $x_*$ and $L - x_*$, interpreted as emergent horizons in a stroboscopic curved space-time [39]. At each Floquet cycle entangled pairs of quasiparticles are created, which accumulate at each of the two peaks, leading to a linear growth of mutual information $I(A;B)$ between the two horizons. We want to understand the robustness of this phenomenology to the introduction of dissipation and thermal initial states. (d) Phase diagram between heating and non-heating phases in the case without dissipation at zero temperature.

volve switching between a uniform Hamiltonian and a spatially modulated Hamiltonian whose modulation is a Sine-Square Deformation (SSD) (see Refs. [33–36] for discussions about SSD systems). A remarkable aspect of this setup is that the dynamics is exactly solvable within the framework of conformal field theory [37]. An exploration of the non-equilibrium dynamics revealed a rich universal phenomenology, where heating and non-heating phases alternate as function of the driving parameters, with universal critical exponents delineating the two phases. Interesting dynamical signatures in observables such as the entanglement entropy, energy density, Loschmidt echo, and dynamical two-point functions were obtained analytically [38,39]. An indepth exploration of the non-ergodic heating regimes unearthed a complex spatial structure: energy evolution was found to be extremely inhomogeneous in space, with two emergent hotspots where energy accumulated with time. Such hotspots were found to share growing entanglement, and were interpreted as black hole horizons in an effective stroboscopic curved space-time [39]. Different extensions of these drives were then investigated, such as quasi-periodic drives [40,41], random drives [42], non-Hermitian drives [43], continuous drives [44], and general inhomogeneous deformations [45,46]. An important question is whether such phenomenology survives the onset of dissipation and whether dissipation suffices to eliminate this "integrable heating", as opposed to usual ergodic heating.

In this paper we focus on the driven-dissipative dynamics of a critical free fermion chain that is periodically driven following the SSD drive protocol and can exchange particles with an external bath. We start with a brief review of the dynamics of periodically driven CFTs with

spatially modulated profiles in Sec. 2, and summarise the main notions that will be used in the rest of the paper to compare with our numerical findings. In Sec. 3 we introduce the setup studied in our paper, and write down the main equations that are numerically integrated to obtain the full stroboscopic dynamics of the correlation matrix in the presence of different types of dissipation. We then introduce initial thermal states in Sec. 4, for which we compute the stroboscopic time evolution of energy analytically using CFT, and then compute numerically the time evolution of the correlation matrix, from which we infer energy density evolution as well as entanglement entropy evolution. We explicitly compare predictions from periodically driven CFTs at finite temperature to results on the fermionic chain, and find that remarkably the CFT predictions about the existence of heating and non-heating phases, as well as the precise scaling across the transition, are still valid for relatively large initial temperatures. The spatial structure inherent to the heating phase of this Floquet drive at zero temperature is also present at finite temperatures, and can be observed both as signatures in energy density and entanglement entropy. We then study the effect of dissipation in Sec. 5, by putting two dissipators at the end of the chain, which can exchange particles with an external bath or that can act as a source of dephasing. In these two driven-dissipative scenarios, energy as well as entanglement increase rapidly, but clear signatures of the emergent horizons are observed in the high-frequency regime, where particles entering the system from the bath get stuck at the first horizon they encounter. Away from the high-frequency limit, though signatures of the horizons are not as easily observable in the energy/particle density, the specific kink structure of the mutual information between the two horizons survives a wide range of dissipation strengths.

## 2   Review of Floquet CFT

The non-equilibrium dynamics of interacting lattice models is in general a hard problem to solve. In this paper we exclusively consider the case of critical one-dimensional lattice models, i.e., those whose long wavelength theory have emergent conformal invariance, and are well-described by a CFT. While their driven lattice counterparts are in general not integrable, here we focus on a class of solvable periodically driven CFTs. To set the stage for the main goals of our work, we first review the physics of periodically driven inhomogeneous CFTs followed by a direct comparison between the analytic predictions of the CFT and numerical results on a driven critical one dimensional lattice model. We consider a (1+1)-dimensional CFT of length $L$ describing gapless excitations of the critical system governed by the inhomogeneous Hamiltonian

$$\mathcal{H} = \int_0^L v(x) T_{00}(x), \tag{1}$$

where $v(x)$ is a smooth and positive deformation profile, and $T_{00}(x)$ is the energy density of the CFT, with $v(x) \equiv v$ being the case of the homogeneous Hamiltonian. Using the tools developed in Ref. [47–50], such an inhomogeneous CFT can be transformed to a homogeneous one via a change of coordinates:

$$f(x) = \int_0^x \mathrm{d}x' \frac{\tilde{v}}{v(x')}, \qquad \frac{1}{\tilde{v}} = \frac{1}{L} \int_0^L \frac{\mathrm{d}x'}{v(x')}. \tag{2}$$

Consequently, the Heisenberg time evolution of any primary field is implemented by the change of coordinate $f(x)$. In the rest of the manuscript, we will set the velocity of gapless quasiparticles $v$ to 1, as well as $\hbar = 1$.

To study the Floquet dynamics of these generically deformed CFTs, we consider a 2-step drive alternating between an inhomogeneous Hamiltonian $\mathcal{H}_0$ with deformation $v_0(x)$ applied

for time $T_0$ and another inhomogeneous Hamiltonian $\mathcal{H}_1$ with deformation $v_1(x)$ applied for time $T_1$ [45,46]. The Floquet unitary is then given by

$$U_F = e^{-i\mathcal{H}_0 T_0} e^{-i\mathcal{H}_1 T_1} \,. \tag{3}$$

It is possible to explicitly derive a change of coordinates encoding the 1-cycle time evolution of any primary field $\phi(x,t) = U_F^\dagger \phi(x) U_F$ of conformal weight $(h, \bar{h})$ [45]. Using this in conjunction with the transformation law of primary fields under a conformal transformation, one obtains the following Floquet time evolution in Heisenberg picture:

$$\phi(x,t) = \left[\frac{\partial \tilde{x}_n^-(x)}{\partial x}\right]^h \left[\frac{\partial \tilde{x}_n^+(x)}{\partial x}\right]^{\bar{h}} \phi(\tilde{x}_n^-(x), \tilde{x}_n^+(x)), \tag{4}$$

where

$$\tilde{x}_n^\mp(x) = f_\pm(\tilde{x}_{n-1}^\mp(x)), \qquad \tilde{x}_0^\mp(x) = x\,, \tag{5}$$

and the transformations $f_\pm$ are defined by

$$f_\pm(x) = f_1^{-1}\left(f_1\left(f_0^{-1}\left(f_0(x) \mp \frac{T_0}{L}\right)\right) \mp \frac{T_1}{L}\right), \tag{6}$$

where $f_0$ and $f_1$ are the quantities specified in (2) for the two Hamiltonians $\mathcal{H}_0$ and $\mathcal{H}_1$.

The full stroboscopic time evolution of any (quasi)-primary field after $n$ Floquet cycles is therefore given by a composition of $n$ 1-cycle maps (6). In general it is not possible to write down a closed form expression for such a composition. However it turns out that the Floquet dynamics is completely encoded by fixed points of $f_\pm^n = f_\pm \circ \ldots \circ f_\pm$: the energy oscillates in a bounded manner if there are no unstable fixed points; if the $n$-cycle map admits any unstable fixed points, $f_\pm^n(x_*^\mp) = x_*^\mp$ and $f_\pm^{n\prime}(x_*^\mp) > 1$, all correlation functions will increase exponentially at such a point, leading to an exponential growth of total energy $E(t = nT) = \int_0^L dx \langle T_{00}(x)\rangle$, where $\langle\ldots\rangle$ denotes average with respect to the time-evolved state $(U_F)^n |\psi_0\rangle$. The associated heating rate is simply given by $\frac{1}{2(T_0+T_1)n}\log(f_\pm^{\prime n}(x_*^\mp))$, where $x_*^\mp$ is the most unstable fixed point of $f_\pm^n$. This increase of total energy is extremely inhomogeneous in space, and restricted to the spatial positions given by the set of unstable fixed points of $f_\pm^n$. The energy decreases exponentially at all other spatial locations. The phase transition between non-heating and heating phases corresponds to stable and unstable fixed points merging into a single tangent point, $f_\pm^{\prime n}(x_*^\mp) = 1$. In general, one can also have Lifhshitz-like transitions between phases with differing number of unstable fixed points which manifest as a change in the kink structure of the entanglement entropy [45]. We note that the phase diagram in the general case can only be obtained iteratively, as it involves finding fixed points of $f_\pm^n$ for any $n \in \mathbb{N}$. The phase transitions as well as boundaries delineating heating phases are closely linked to parametric resonance and Arnold tongues [51].

For general deformations of the Hamiltonian density one expects to have mostly heating in the infinite time limit, with non-extended non-heating "phases". In contrast, considering a deformation that involves a single Fourier mode leads to an exact extended non-heating phase. We summarize the results for the particular case where the Hamiltonian deformations $v_0(x)$ and $v_1(x)$ only consist of a single Fourier mode, i.e.,

$$v_i(x) = \alpha_i + \beta_i \cos\left(\frac{2\pi x}{L}\right) + \gamma_i \sin\left(\frac{2\pi x}{L}\right), \quad i \in \{0, 1\}. \tag{7}$$

These two deformed Hamiltonians belong to the $\mathfrak{sl}(2)$ subalgebra of the infinite dimensional Virasoro algebra, spanned by the Virasoro generators $\{L_1, L_{-1}, L_0, \overline{L}_1, \overline{L}_{-1}, \overline{L}_0\}$. This particularly

simple algebraic structure implies that on the complex plane $z = e^{2\pi i x / L}$ the coordinate transformation encoding the 1-cycle evolution, $\tilde{z}$, is an invertible Möbius transformation [52, 53]

$$\tilde{z} = e^{\frac{2\pi i f_+(x)}{L}} = \frac{az + b}{cz + d}, \quad \tilde{\bar{z}} = e^{\frac{2\pi i f_-(x)}{L}} = \frac{a\bar{z} + b}{c\bar{z} + d}. \tag{8}$$

This fact greatly simplifies the Floquet dynamics compared to the generic case involving the full Virasoro algebra described above, as the composition of Möbius transformations has a $SL(2, \mathbb{C})$ group structure, and just amounts to matrix multiplication. The $n$-cycle map is then given by

$$\tilde{z}_n = e^{\frac{2\pi i f_+^n(x)}{L}} = \frac{a_n z + b_n}{c_n z + d_n}, \quad \tilde{\bar{z}}_n = e^{\frac{2\pi i f_-^n(x)}{L}} = \frac{a_n \bar{z} + b_n}{c_n \bar{z} + d_n}, \tag{9}$$

where $\begin{pmatrix} a_n & b_n \\ c_n & d_n \end{pmatrix} = \begin{pmatrix} a & b \\ c & d \end{pmatrix}^n$. The Floquet dynamics is then fully classified by the sign of $\Delta = (a + d)^2 - 4$ :

1. $\Delta > 0$: *Elliptic* Möbius transformation corresponding to a non-heating phase.

2. $\Delta < 0$: *Hyperbolic* Möbius transformation corresponding to a heating phase.

3. $\Delta = 0$: *Parabolic* Möbius transformation corresponding to the phase transition.

We note that elliptic transformations have no fixed points, while the hyperbolic classes have one stable and one unstable fixed point, that we denote in the complex plane by $\gamma_1$ and $\gamma_2$, given explicitly by

$$\begin{cases} \gamma_1 = \frac{a - d - \sqrt{(a-d)^2 + 4bc}}{2c}, \\ \gamma_2 = \frac{a - d + \sqrt{(a-d)^2 + 4bc}}{2c}. \end{cases} \tag{10}$$

Assuming $\gamma_2$ is the unstable fixed point, then energy density $\langle T_{00}(x) \rangle$ will concentrate exponentially at $x_* = \frac{L}{2\pi i} \log(\gamma_2)$ and $L - x_* = -\frac{L}{2\pi i} \log(\gamma_2)$, and decay exponentially in time at any other point.

For concreteness we consider a 2-step drive between the homogeneous Hamiltonian $\mathcal{H}_0$ and the SSD Hamiltonian $\mathcal{H}_1$, given by the deformation $v(x) = 1 - \cos\left(\frac{2\pi x}{L}\right) = 2\sin^2\left(\frac{\pi x}{L}\right)$, as first studied in Ref. [37]. In this case, the 1-cycle transformation reads

$$\tilde{z} = \frac{\left(1 + \frac{i\pi T_1}{L}\right) e^{\frac{i\pi T_0}{L}} z - \frac{i\pi T_1}{L} e^{-\frac{i\pi T_0}{L}}}{\frac{i\pi T_1}{L} e^{\frac{i\pi T_0}{L}} z + \left(1 - \frac{i\pi T_1}{L}\right) e^{-\frac{i\pi T_0}{L}}}. \tag{11}$$

Correspondingly,

$$\Delta = \left[1 - \left(\frac{\pi T_1}{L}\right)^2\right] \sin^2\left(\frac{\pi T_0}{L}\right) + \frac{\pi T_1}{L} \sin\left(\frac{2\pi T_0}{L}\right), \tag{12}$$

and we see that the phase diagram is now fixed by the sign of $\Delta$, see Fig. 1(d). Specifically, for the initial state $|\psi_0\rangle$ (the ground state of $\mathcal{H}_0$) and open boundary conditions, the total energy is given by [38, 39]

$$E(t = nT) = \int_0^L \mathrm{d}x \langle T_{00}(x) \rangle = \frac{2\pi}{L} \frac{c}{16} \frac{a_n d_n + b_n c_n}{a_n d_n - b_n c_n}, \tag{13}$$

where $c$ is the central charge of the theory. $E(t)$ oscillates in the non-heating phase while it grows exponentially in the heating phase. The parameter which describes both the periodicity in the non-heating phase and the heating rate in the heating phase is

$$\eta = \frac{a + d + \sqrt{(a-d)^2 + 4bc}}{a + d - \sqrt{(a-d)^2 + 4bc}}, \tag{14}$$

such that the periodicity is simply $T_E = \frac{T_0 + T_1}{2\pi |\log(\eta)|}$, while in the heating phase, the heating rate is given by $T_E^{-1}$. We note that $\eta$ is on the unit circle in the non-heating phase, $|\eta| = 1$, while it is a positive real number in the heating phase, which leads to the two different stroboscopic time evolutions of both phases. For an initial primary state $|\Phi\rangle$ of conformal weight $\Delta$ and periodic boundary conditions, one can show that the prefactor $\frac{c}{16}$ in Eq. (13) gets replaced by $2\Delta$ [44]. The stroboscopic effective Hamiltonian can be shown to be $\mathcal{H}_F = \mathfrak{a}L_0 + \mathfrak{b}L_{-1} + \mathfrak{c}L_1$, wherein the phase boundaries are delineated by the sign of the quadratic Casimir invariant of $\mathfrak{sl}(2)$, $c^{(2)} = \mathfrak{a}^2 - 4\mathfrak{b}\mathfrak{c}$, as shown in Ref. [39]. Interestingly, the propagation of gapless quasiparticles in an inhomogeneous CFT can be understood as light-like geodesics in a curved space-time [54] specified by the metric $ds^2 = dx^2 - v(x)^2 dt^2$. Consequently, the stroboscopic propagation of quasiparticles under the SSD drive can be viewed as a propagation in a curved space-time containing two black hole horizons at the afore-mentioned positions $x_*$ and $L - x_*$, corresponding to the unstable fixed points where energy and quasiparticles accumulate.

The entanglement entropy $S_A(t)$ can also be computed starting from $|\psi_0\rangle$ with open boundary conditions [55]. In the non-heating phase, it simply oscillates with a periodicity $T_E$, while in the heating phase, as long as the block $A$ contains one and only one of the two horizons $x_*$ and $L - x_*$, $S_A(t)$ grows linearly in time (with a universal part $\sim -\frac{c}{6} n \log(\eta)$ where $\sim$ denotes only up to non-universal terms). From a quasiparticle perspective, at each Floquet cycle entangled pairs of quasiparticles are created and accumulate at one horizon each. As a consequence, the growth of entanglement must be shared only between the two horizons, as illustrated on Fig. 1(c). This is quantified by the mutual information $I(A; B) = S_A + S_B - S_{AB}$, which grows linearly with the number of cycles only if $x_* \in A$ and $L - x_* \in B$ or vice versa, $I(A; B) \sim -\frac{c}{3} n \log(\eta)$ [38]. Hence, from the curved space-time viewpoint, the inhomogeneous heating phase at stroboscopic times manifests two entangled black hole horizons absorbing all energy.

To summarise, this class of exactly solvable periodically driven CFTs provides us with a set of exact results characterized solely by the central charge $c$ of the theory for the stroboscopic time evolution of a multitude of physical observables. These predictions were found to be in remarkable agreement with numerical results obtained in one-dimensional critical lattice models [39]. It is important to note that as the CFT is a continuum theory, it has an infinite number of degrees of freedom leading to unbounded growth of energy and entanglement entropy in the heating phase. On the other hand, in a finite size lattice system due to the finite dimensionality of the Hilbert space, CFT predictions are typically valid only up to to a cutoff timescale, for example, $10 - 20$ Floquet cycles in the heating phase [38], while in the non-heating phase the CFT results remain valid for much larger timescales as the entanglement entropy and energy time evolutions remain bounded in the CFT description.

# 3  Dissipative dynamics

In the previous section, we discussed how heating in a generic class of critical driven closed systems emerges via with formation of entangled energy hotspots. We now explore whether this rich phenomenology survives in an open system setting, a situation of great relevance to

experiments where dissipation is ubiquitous. Typically, the study of a dissipative and driven interacting lattice model at criticality requires highly complex computational tools which might be poorly convergent in the long time limit. However, here we can harness the fact that the main physical features are universal and solely characterized by the central charge of the critical lattice model to simplify our task of the study of the dissipative system. Hence, in this section, as a representative example, we consider a system of free fermions hopping on a one-dimensional lattice of length $L$ at half-filling, whose low energy theory is a $c = 1$ free boson CFT. This model permits an exact derivation of the time evolution of the system in the presence of both dissipation and drive. The Hamiltonians corresponding to the two-step drive in Eq. (3) are given by

$$
\begin{aligned}
H_0 &= \frac{1}{2}\sum_{i=1}^{L-1} c_i^\dagger c_{i+1} + h.c.\,, \\
H_1 &= \sum_{i=1}^{L-1} \sin^2\left(\frac{\pi i}{L}\right) c_i^\dagger c_{i+1} + h.c.
\end{aligned}
\tag{15}
$$

The dynamics of the system density matrix $\rho$ is governed by the Gorini-Kossakowski-Sudarshan-Lindblad master equation [56–61]

$$
\frac{\partial \rho}{\partial t} = -i[H(t),\rho] + \sum_\mu \left(2L_\mu \rho L_\mu^\dagger - \{L_\mu^\dagger L_\mu, \rho\}\right).
\tag{16}
$$

For the situation where the Hamiltonian $H(t)$ and the bath operators $L_\mu$ are respectively quadratic and linear in the fermionic operators ($c$ and $c^\dagger$), a general solution can be exactly obtained [62].

We first map the fermionic operators to Hermitian Majorana operators

$$
w_{2m-1} = c_m^\dagger + c_m\,, \quad w_{2m} = i(c_m - c_m^\dagger)\,,
\tag{17}
$$

which satisfy the anti-commutation relations

$$
\{w_j, w_k\} = 2\delta_{j,k}\,.
\tag{18}
$$

In the Majorana representation, the Hamiltonian and the bath operators can be expressed as

$$
\begin{aligned}
H &= \sum_{lm} w_l H_{lm} w_m\,, \\
L_\mu &= \sum_n l_{\mu,n} w_n\,,
\end{aligned}
\tag{19}
$$

where $H_{lm}$ is a $2L$ by $2L$ anti-symmetric matrix. In this work, we consider the following bath operators attached at the two ends of the chain as shown in Fig. 1(a):

$$
L_{L/R} = \Gamma_+^{L/R} c_{L/R}^\dagger + \Gamma_-^{L/R} c_{L/R}\,,
\tag{20}
$$

where $L$ ($R$) refers to the site on the left (right) edge of the chain. For this work, we fix $\Gamma^L = \Gamma^R$ and define

$$
\Gamma_+ = \gamma\,, \quad \Gamma_- = \mathcal{R}\gamma\,.
\tag{21}
$$

In a non-interacting system, all observables can be obtained from the correlation matrix

$$
C_{lm} = \mathrm{tr}(w_l w_m \rho)\,.
\tag{22}
$$

Using the anti-commutativity of the Majorana operators (18) and the Lindblad equation (16), it can be shown that the correlation matrix obeys [63]

$$\frac{dC}{dt} = -4C(t)\left[iH(t) + iH(t)^T + M_\mathrm{r} + M_\mathrm{r}^T\right] - 8iM_\mathrm{i}\,, \tag{23}$$

where $M_\mathrm{r}$ ($M_\mathrm{i}$) is the real (imaginary) part of the the matrix $M_{ij} = \sum_\mu l_{\mu,i} l_{\mu,j}^*$. We now use the fact that the Floquet Hamiltonian $H(t)$ is piecewise constant in time, such that Eq. (23) is a Lyapunov matrix ordinary differential equation of the form $\dot{C} = -XC(t) - C(t)X^T - iY$, whose explicit solution takes the closed form [64]

$$C_{il}(t) = \sum_{j,k} V_{ij}\left[\left(e^{t(\alpha_j + \beta_k)}\right)\left(V^{-1}C(t_0)(W^\dagger)^{-1}\right)_{jk} + \left(\int_{t_0}^t ds\, e^{(t-s)(\alpha_j + \beta_k)}\right)\left(V^{-1}(-iY)(W^\dagger)^{-1}\right)_{jk}\right]W_{kl}^\dagger\,, \tag{24}$$

where $C(t_0)$ is the initial correlation matrix, $V$ and $W$ are the unitaries diagonalizing $-X$ and $-X^T$, and $\alpha_i$, $\beta_i$ are their respective eigenvalues. The full stroboscopic time evolution $C(n(T_0 + T_1))$ can be obtained numerically via a sequential evolution by resetting the initial condition to $C(nT_1 + (n-1)T_0)$.

This analysis can be extended to include dissipation quadratic in fermion operators, such as on-site dephasing, for which $L_\mu = \sqrt{\gamma_\mu} c_\mu^\dagger c_\mu$. By taking expectation values of Eq. (16) one finds the evolution equation for the (now complex) correlation matrix $\Gamma_{ij}(t) = \mathrm{Tr}\{\rho(t)c_i^\dagger c_j\}$ to be of the form

$$\partial_t \Gamma_{ij}(t) = i\left[h^T(t), \Gamma(t)\right]_{ij} + \sum_\mu \gamma_\mu (2\delta_{ij}\delta_{\mu i} - \delta_{\mu i} - \delta_{\mu j})\Gamma_{ij}(t)\,, \tag{25}$$

where $h(t)$ is defined by $H = \sum_{ij} h_{ij}(t)c_i^\dagger c_j$. Closed equations similar to Eq. (25) can be easily integrated numerically, and hold for generic dephasing terms as long as the jump operators are hermitian. As before, the correlation matrix contains enough information to compute quantities such as the energy density. However, in contrast to single particle loss and gain, the resulting density matrix is in general non-gaussian, which prevents us from directly computing entanglement entropies [65].

Dephasing is also often generated by unitary evolution in the presence of white noise [66], or in quantum state diffusion models [65]. Here we take the former approach by adding an additional Hamiltonian term $V_i = \xi_i(t)n_i$, where $\xi_i(t)$ is gaussian white noise whose variance is fixed by the dephasing strength via $\overline{\xi_i(t)\xi_i(t')} = \delta(t-t')\gamma_i$. After averaging over different noise realizations the unitary dynamics is equivalent to Eq. (16) with $L_\mu = \sqrt{\gamma_\mu} c_\mu^\dagger c_\mu$, which can be seen by averaging over the trotterized equations of motion for the density matrix. This endows the dephasing with a physical interpretation and allows us to study the entanglement dynamics within each noise realization. The average dynamics of the entanglement entropy $\overline{S_A(t)} = -\overline{\mathrm{Tr}\rho_A \log \rho_A} \neq -\mathrm{Tr}\overline{\rho}_A \log \overline{\rho}_A$ generally differs from the entanglement of the average density matrix, and can exhibit more interesting features such as entanglement phase transitions [67]. In the following we will focus on the dynamics of $\overline{S_A(t)}$ and the energy density for boundary noise where

$$L_{L,R} = \sqrt{\gamma_{L,R}} c_{L,R}^\dagger c_{L,R} = L_{L,R}^\dagger\,. \tag{26}$$

## 4 Thermal Initial States

Before investigating the impact of dissipation on the driven lattice, we first address whether the heating and non-heating phases studied in [37–39, 45] are robust to thermal initial states instead of pure initial states, both within the CFT description and on the lattice.

## 4.1 Thermal Floquet CFT

In this section we provide an analytical expression for the stroboscopic time evolution of the total energy $E(t)$ after $n$-cycles of the SSD Floquet drive (see Sec. 2), starting from an initial thermal state at temperature $\beta^{-1}$. This entails the computation of the following:

$$E(t) = \frac{2\pi}{L}\text{Tr}(e^{-\beta\mathcal{H}_0}(U_F^n)^\dagger(L_0 + \bar{L}_0)U_F^n), \tag{27}$$

where $U_F = e^{-i\mathcal{H}_0 T_0}e^{-i\mathcal{H}_1 T_1}$. We provide two alternative routes to compute $E(t)$. The first approach makes use of the underlying $\mathfrak{su}(1,1)$ algebra spanned by $\{L_0, L_{-1}, L_1\}$, while the second approach relies on diffeomorphisms of the circle.

We first evaluate the stroboscopic time evolution operator

$$(U_F^n)^\dagger L_0 U_F^n = e^{in(T_0+T_1)\mathcal{H}_F}L_0 e^{-in(T_0+T_1)\mathcal{H}_F}, \tag{28}$$

with the Floquet Hamiltonian assuming the form

$$\mathcal{H}_F = [\mathfrak{a}L_0 + \mathfrak{b}L_{-1} + \mathfrak{c}L_1] + \text{anti-holomorphic part}. \tag{29}$$

The coefficients $\mathfrak{a}, \mathfrak{b}, \mathfrak{c}$ were evaluated explicitly in Ref. [39]. Concretely, for the SSD drive protocol, the effective Hamiltonian reads

$$\begin{aligned}\mathcal{H}_F = &\frac{i}{T_0+T_1}\frac{\log\eta}{(\gamma_1+\gamma_2)\gamma_1-\gamma_2}\left[L_0 - \frac{1}{2}(\gamma_1\gamma_2+1)(L_1+L_{-1}) - \frac{1}{2}(\gamma_1\gamma_2-1)(L_{-1}-L_1)\right]\\ &+\frac{i}{T_0+T_1}\frac{\log\eta}{(\gamma_1+\gamma_2)\gamma_1-\gamma_2}\left[\bar{L}_0 - \frac{1}{2}(\gamma_1\gamma_2+1)(\bar{L}_1+\bar{L}_{-1}) + \frac{1}{2}(\gamma_1\gamma_2-1)(\bar{L}_{-1}-\bar{L}_1)\right].\end{aligned} \tag{30}$$

where the fixed points $\gamma_1, \gamma_2$ and the multiplier $\eta$ are given by (10) and (14) respectively. The time evolution of the operator $L_0$ in the Heisenberg picture takes the form

$$e^{in(T_0+T_1)\mathcal{H}_F}L_0 e^{-in(T_0+T_1)\mathcal{H}_F} = \theta_1 L_0 + \theta_2 L_1 + \theta_3 L_{-1}. \tag{31}$$

Using the fact that only $L_0$ has a non-zero expectation in a thermal state, we obtain

$$\text{Tr}[(L_0 + L_1 + L_{-1})e^{-\beta\mathcal{H}_0}] = \text{Tr}[L_0 e^{-\beta\mathcal{H}_0}]. \tag{32}$$

Note that the non-zero modes of the stress tensor do not contribute since $L_{\pm 1}$ acting on the ket (or bra) creates a descendant state of a different level, which is orthogonal to the bra (or ket). From this argument, it is clear that $\langle L_1\rangle_\beta = \langle L_{-1}\rangle_\beta = 0$, thus we simply need to evaluate $\theta_1$ in (31). To do so, we make use of the non-unitary $2\times 2$ representation of the $\mathfrak{su}(1,1)$ algebra, given by

$$L_0\Big|_{2\times 2} = \begin{pmatrix} -1/2 & 0 \\ 0 & 1/2 \end{pmatrix}, \qquad L_{-1}\Big|_{2\times 2} = \begin{pmatrix} 0 & 0 \\ -1 & 0 \end{pmatrix}, \qquad L_1\Big|_{2\times 2} = \begin{pmatrix} 0 & 1 \\ 0 & 0 \end{pmatrix}. \tag{33}$$

We thus deduce that

$$\langle e^{i\mathcal{H}_F t}L_0 e^{-i\mathcal{H}_F t}\rangle_\beta = \theta_1\langle L_0\rangle_\beta = \left(\frac{\mathfrak{a}^2 - 4\mathfrak{b}\mathfrak{c}\cos\left[\sqrt{\mathfrak{a}^2-4\mathfrak{b}\mathfrak{c}}\frac{2\pi t}{L}\right]}{\mathfrak{a}^2 - 4\mathfrak{b}\mathfrak{c}}\right)\langle L_0\rangle_\beta, \tag{34}$$

The remaining thermal expectation value is a time-independent equilibrium one, thus the full stroboscopic time evolution is encoded in $\theta_1$. Restricting to the $c = 1$ free boson CFT we can use the following standard result on the torus for a Luttinger liquid at $K = 1$,

$$\langle L_0\rangle_\beta = -\frac{L}{4\pi}\frac{\partial\log\Theta(\beta/L)}{\partial\beta} + \sum_{m>0}\frac{m}{e^{2\pi m\beta/L}-1}, \tag{35}$$

where $\Theta$ is the Siegel theta function, explicitly defined as

$$\Theta = \sum_{m,\omega\in\mathbb{Z}} \exp\left[-\frac{\pi\beta}{L}\left(\frac{m^2}{2} + 2\omega^2\right)\right]. \tag{36}$$

Using the above results in (30), we obtain the following general result for the stroboscopic energy evolution after $n$ cycles,

$$E(n) = \frac{\gamma_1\gamma_2 - 4\cosh\left(\log\eta\,\frac{\sqrt{\gamma_1\gamma_2}\sqrt{\gamma_1\gamma_2-4}}{\gamma_1-\gamma_2}n\right)}{\gamma_1\gamma_2 - 4}\left[-\frac{L}{4\pi}\frac{\partial\log\Theta(\beta/L)}{\partial\beta} + \sum_{m>0}\frac{m}{e^{2\pi m\beta/L} - 1}\right]. \tag{37}$$

The principal effect of temperature manifests via an overall temperature dependent prefactor stemming from the equilibrium thermal expectation value $\langle L_0\rangle_\beta$. Consequently, both the periodicitiy of the energy oscillations and the heating rate remain unaltered. Though this factor depends on the specific theory at hand, temperature effectively decreases the energy amplitudes in both phases. The universal critical exponent characterising the non-heating-to-heating phase transition is still $\frac{1}{2}$, as the order parameters (the periodicity and the heating rate) are identical with respect to the zero temperature case.

We note that the above derivation only holds for deformation profiles that belong to the $\mathfrak{sl}(2)$ subalgebra of the full Virasoro algebra. We can also derive the finite-temperature evolution of the energy-momentum tensor after the 2-step Floquet drive for generic deformations based on the geometric approach from Ref. [49]. In this case, the evolution of the holomorphic part of the stress tensor is given by

$$\langle T(x,t)\rangle_\beta = \left(\frac{\partial\tilde{x}_n^-}{\partial x}\right)^2\langle T(\tilde{x}_n^-(x))\rangle_\beta - \frac{c}{24\pi}\{\tilde{x}_n^-, x\}, \tag{38}$$

where $\tilde{x}_n^-(x)$ is given by (5), and $\{\tilde{x}_n^-, x\}$ is the Schwarzian derivative of $x_n^-(x)$. Combining the holomorphic and anti-holomorphic parts of the stress tensor, we conclude that the energy density is

$$E(x,t) = \left[\left(\frac{\partial\tilde{x}_n^-}{\partial x}\right)^2 + \left(\frac{\partial\tilde{x}_n^+}{\partial x}\right)^2\right]\langle T\rangle_\beta - \frac{c}{24\pi}[\{\tilde{x}_n^-, x\} + \{\tilde{x}_n^+, x\}]. \tag{39}$$

We can then use the fact that the equilibrium one-point function of the stress tensor $\langle T(\tilde{x}_n^-(x))\rangle_\beta = \langle \bar{T}(\tilde{x}_n^+(x))\rangle_\beta = \langle T\rangle_\beta$ is independent of $x$, and is simply given by the derivative of the partition function on the torus with respect to $\beta$. For a $c = 1$ CFT corresponding to the $K = 1$ Luttinger liquid, we find

$$\langle T\rangle_\beta = \langle\bar{T}\rangle_\beta = -\frac{\partial\log Z}{\partial\beta}, \tag{40}$$

with the partition function of the free boson at $K = 1$ given by

$$Z(\beta) = \frac{1}{|\eta(i\beta/L)|^2}\sum_{m,w\in\mathbb{Z}}\exp\left[-\pi\frac{\beta}{L}\left(\frac{m^2}{2} + 2w^2\right)\right] = \frac{\Theta(\beta/L)}{|\eta(i\beta/L)|^2}, \tag{41}$$

where $\eta$ is the Dedekind eta function. This formula enables one (by integrating over space) to obtain $E(t)$,

$$E(t) = \int_0^L E(x,t)\mathrm{d}x. \tag{42}$$

For the case of the SSD drive, the 1-cycle diffeomorphisms (6) are given by

$$f_\pm(x) = f^{-1}(f(x) \mp T_0) \mp T_1 = \frac{L}{\pi} \arctan\left[\tan\left(\frac{\pi x}{L}\right) \mp 2\pi \frac{T_0}{L}\right] \mp \frac{T_1}{L}. \tag{43}$$

This approach thus leads to the stroboscopic evolution of energy density and total energy after $n$-cycles for general deformation profiles. However, the result does not have a closed form and must be iterated for each Floquet cycle. In the rest of the work, we concentrate on the SSD drive protocol for which we have a closed form expression for (37).

## 4.2 Thermal initial states on the lattice

We now turn to lattice calculations at finite temperature. We consider the following initial thermal correlation matrix

$$C_{ij} = \frac{1}{\text{tr}(e^{-\beta H_0})} \text{tr}(c_i^\dagger c_j e^{-\beta H_0}), \tag{44}$$

where $H_0$ is the uniform chain defined in Eq. (15). This expression reduces to evaluating

$$C_{ij} = \sum_k U_{ki}^* U_{kj} \langle n_k \rangle, \tag{45}$$

where $U$ is the unitary diagonalizing the matrix $h_0$. In the large $N = \sum_k \langle n_k \rangle$ limit, we can use the Fermi-Dirac distribution

$$\langle n_k \rangle = \frac{1}{e^{\beta(\epsilon_k - \mu)} + 1}. \tag{46}$$

The time evolution of the correlation matrix is then obtained by solving Eq. (23) numerically for zero dissipation, which then reduces to the Heisenberg equation.

We extended previous CFT results at $\beta^{-1} = 0$ at thermal initial states in Sec. 4.1, in the case of a $c = 1$ compactified free boson CFT on a radius $R = 2$ (or equivalently with a the Luttinger parameter $K = 1$), which describes the low-energy dynamics of our lattice model. We conclude that the physical behaviour predicted by thermal Floquet CFT in the heating and non-heating phases does not depend crucially on the choice of (pure) initial state, which simply changes the amplitude of the total energy evolution, c.f. (37). In order to verify the CFT predictions, we revert to the lattice model and compute the energy, $E(t) = \sum_i \langle c_{i+1}^\dagger c_i \rangle + h.c.$ Our results for different temperatures and a comparison with the CFT predictions are shown in Fig. 2. In the heating phase, an exponential growth of energy with higher energy absorption at finite temperature is seen, as predicted by the finite temperature CFT predictions, that agree for a few Floquet cycles with the lattice calculations. Surprisingly, the actual heating rate on the lattice compared to the one predicted by CFT, $\frac{2\pi}{T_0 + T_1}|\log(\eta)|$, slightly decreases with increasing temperature. The non-heating phase persists: the periodicity, formally defined as the inverse of the heating rate of the heating phase, $T_E = \frac{T_0 + T_1}{2\pi|\log(\eta)|}$, grows with temperature in the same way that the heating rate decreases with temperature in the heating phase, while the amplitude of the oscillations gets larger, as predicted from CFT. From a CFT perspective, the change in periodicity and heating rate cannot be explained by the introduction of finite temperature, as seen explicitly from (37). This effect is a consequence of the full cosine dispersion of the lattice Hamiltonian, as taking higher initial temperatures will imply access to high-energy states away from the linearized regime around the Fermi points. We finally note that at long times on the lattice in the non-heating phase, the energy oscillations eventually decay, as shown in Fig. 2(c-d), while CFT predicts persistent oscillations up to infinite times. The speed of such a decay depends non-universally on the initial temperature $\beta^{-1}$ as well as the choice of driving parameters $(T_0/L, T_1/L)$. However it does not depend on the choice of system size $L$, for fixed $T_0/L$ and $T_1/L$.

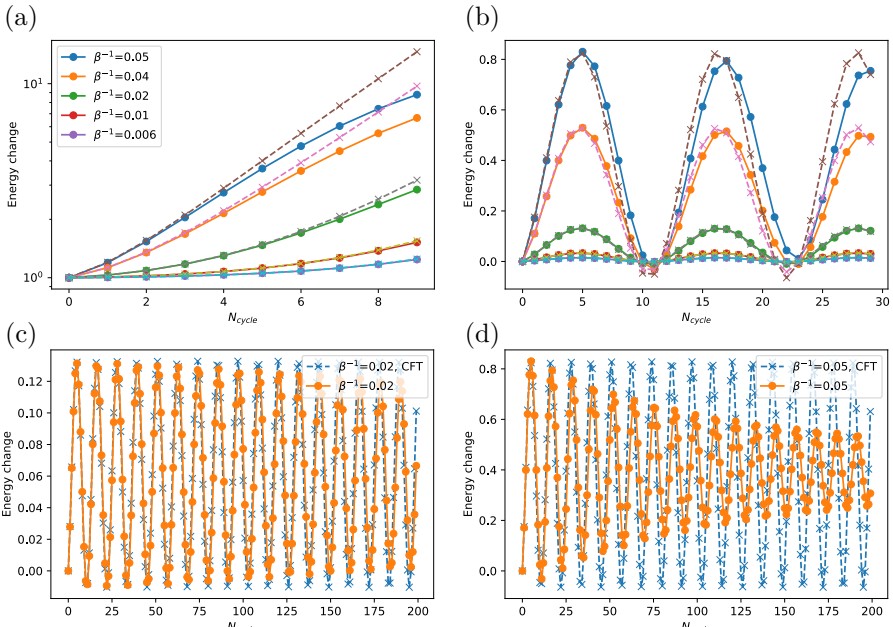

Figure 2: Energy change $E(t) - E(0)$ in heating phase (a) and non-heating phase (b), for different initial temperatures, for a one dimensional chain of length $L = 1000$ with periodic boundary conditions and $|T_0/L| = |T_1/L| = 0.05$. Comparison with the CFT time evolution at finite temperature is shown (dashed lines). The long time limit of the energy evolution is shown in the non-heating phase on the lattice (orange), compared to the CFT predictions (blue), both at initial temperature $\beta^{-1} = 0.02$ (c) and $\beta^{-1} = 0.05$ (d).

To study the robustness of the phase diagram to temperature, we plot $E(t = 10T)$ as a function of the driving parameters $T_0$ and $T_1$, as shown in Fig. 3. At zero temperature, this approximates well the analytically obtained phase diagram of the CFT. At $\beta^{-1} = 0.1$ (to be compared with the cosine bandwidth of 2), the phase diagram remains unaffected and the transition between oscillating and exponentially growing total energy is still clear. This observation is consistent with the expectations from the CFT: in principle, the long time Floquet dynamics should be independent of the initial state as it only involves time evolution of operators in Heisenberg picture, such that both heating and non-heating phases should remain well-defined. This conclusion is further strengthened by the scaling behaviour of the order parameter $\frac{T_0 + T_1}{2\pi |\log(\eta)|}$, the (pseudo-)periodicity of the energy $E(t)$. For pure states, CFT predicts a divergence of this order parameter with a critical exponent $\frac{1}{2}$ [37] as one approaches the boundary to the heating phase. In Fig. 4, we plot the scaling behaviour of the periodicity of the energy as one approaches the heating phase. At zero temperature, the lattice numerics are well-fitted by the CFT predictions. At finite temperature, though the periodicity is modified by temperature as we have observed in Fig. 2, the scaling across the phase transition remains the same and we can still extract a critical exponent of $\frac{1}{2}$ [37], as predicted by thermal Floquet CFT. This indicates that the transition is robust to thermal states as long as $\beta^{-1} < 0.1$, after which broadening effects stemming from the full lattice dispersion become relevant, as observed on Fig. 3.

The hallmark of the heating phase is the emergence of entangled black-hole horizons, where the mutual information $I(A, B)$ between the two horizons $x_*$ and $L - x_*$ grows linearly in time, i.e., $I(A, B) \sim \frac{1}{3} \log(\eta) n$, with $n$ the cycle number, provided $A$ and $B$ both contain one of the two horizons. Using Peschel's method for non-interacting lattice systems [68], we extract

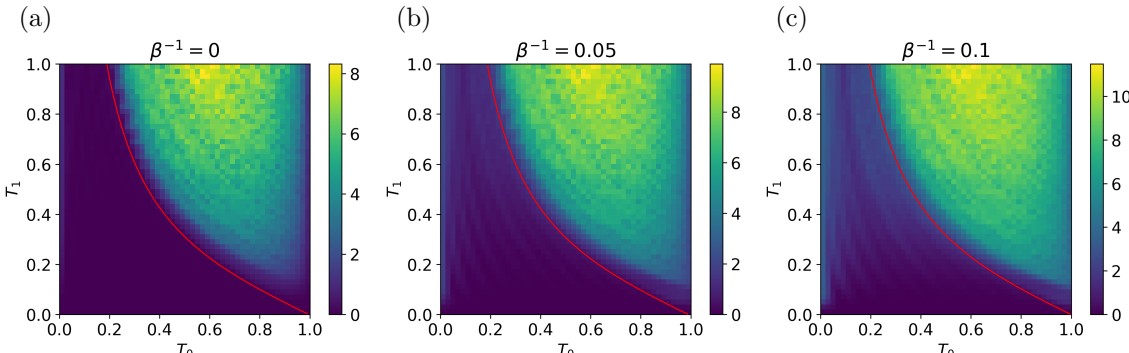

Figure 3: Total energy after 10 cycles as a function of $T_0$ and $T_1$ for $L = 200$, for (a) $\beta^{-1} = 0$, (b) $\beta^{-1} = 0.05$, (c) $\beta^{-1} = 0.1$. The phase boundary between heating and non-heating phases predicted by CFT (at finite and zero temperature) is shown in red and given by the solution of $\Delta = 0$ in Eq. (12). We note that a higher number of Floquet cycles leaves the phase diagram qualitatively unchanged, although the resulting energy deviate from CFT predictions in the heating phase at non-zero initial temperature.

the entanglement entropy from the correlation matrix $C_{ij}$ of the lattice problem. The stroboscopic evolution of the mutual information for $A = [0, x)$ and $B = (x, L]$ for $n = 15$ Floquet cycles with the concomitant energy density $E(x, t)$ are shown in Figs. 5(a) and (b) (heating phase away from the high-frequency limit). At zero temperature, the mutual information displays a clear kink structure at $x_*$ and $L - x_*$, with a linear growth of mutual information if $x \in (x_*, L - x_*)$, and saturation to a constant otherwise. We find that this spatial structure of entanglement is robust to the introduction of initial temperatures, as long as $\beta^{-1} \approx 0.05$, after which the emergent entanglement structure starts to break down, and the energy horizons disappear. We note that this regime of driving parameters corresponds to large micromotion of the order of the system size, such that deviations from the linear dispersion are crucial. We conclude that the CFT predictions for the phase diagram and the structure characterising the heating-to-non-heating transition in the driven lattice model are robust to the introduction of temperatures up to $\beta^{-1} \sim 0.05$.

## 5 Effect of Dissipation

Whether coherent non-equilibrium dynamics remains stable to external baths is an interesting —and in general open— question. The difficulty in open systems stems from the fact that the system can now exchange energy and information with a dissipative environment, which can irreversibly alter fragile quantum states. Typically, one can expect that a contact with the environment acts as a mitigating influence on the energy absorbed by the system from the drive, thereby stabilising non-heating phases in larger parts of the phase diagram. To test this heuristic picture, we consider boundary dissipation in the finite chain as described in Sec. 3. We will not be interested in the steady state reached by the system at large times, but rather in the robustness of the features of the heating phase to dissipation, as well as the transition between different phases at short times, of the order of tens of Floquet cycles.

We first consider the case where the driven free fermion chain exchanges particles with an external bath at its boundaries. As discussed in Sec. 3, all observables can be computed from the correlation matrix (22). The time dependent correlation matrix for dissipative particle exchange is explicitly given by Eq. (24), for a piecewise constant Floquet Hamiltonian $H(t)$.

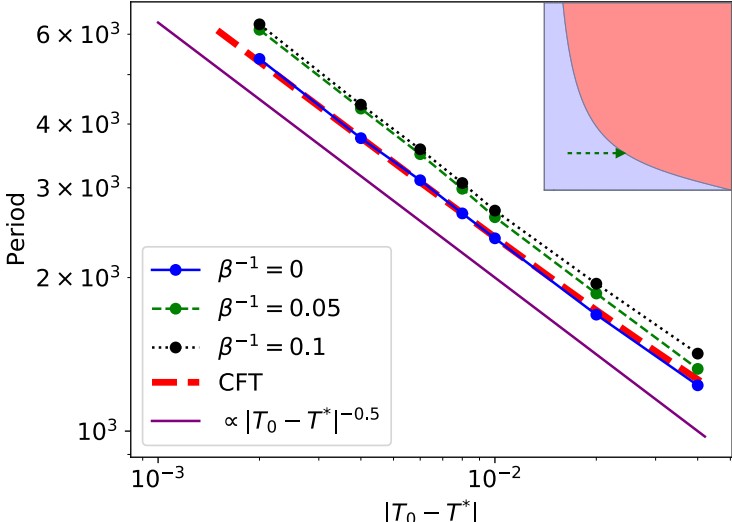

Figure 4: Scaling of the periodicity of total energy $E(t)$ in the non-heating phase when approaching phase transition at $T_0 = T_*$, for $T_1/L = 0.05$, and for different initial states $\beta^{-1}$. Explicit comparison with the CFT scaling (which is independent of temperature) is shown. While the periodicity changes as we increase initial temperature, its scaling near the phase transition agrees with the CFT prediction giving a critical exponent $\frac{1}{2}$ for any initial temperature.

As a first step, we consider the high-frequency regime of the drive, i.e. $|T_0| + |T_1| \ll L$. For $T_0, T_1 > 0$, this corresponds to the non-heating phase in the dissipationless case, as seen in Fig. 1(d). Dissipation washes out the oscillating structure of energy and entanglement entropy of the non-heating phase. We will therefore focus our analysis on the heating phase. Note that addition of temporal disorder already erases the non-heating phase in the dissipationless setting [42]. Given the periodicity of the phase diagram along the $\frac{T_0}{L}$ axis, we can also explore a heating phase in the high-frequency limit if $T_0 < 0$. This is equivalent to switching the sign of the homogeneous Hamiltonian. In this case the quasiparticles propagating with local velocities $v_0(x)$ for time $T_0$ and $v_1(x)$ for time $T_1$, change their direction between the two steps of the drive, and the micro-motion of left (right) movers oscillates around $x_*$ ($L - x_*$). This regime will be particularly resistant to the introduction of dissipation at the two edges of the chain as due to reduced micromotion, the energy horizons survive not only at stroboscopic times but all times. Consequently, no information can propagate from one edge of the system to the other. Therefore, adding or removing particles (depending on the ratio $\mathcal{R}$ between $\Gamma_+^{L/R}$ and $\Gamma_-^{L/R}$) at the edges of the chain does not affect the particle density between $x_*$ and $L - x_*$, which remains pinned at $\frac{1}{2}$, as seen in Fig. 6(c–d) .

Energy accumulation is seen in the regions $[0, x_*]$ and $[L - x_*, L]$ for non-zero dissipation strength, see Fig. 6(b). Although the energy density ultimately becomes larger at the edges of the chain, the horizons act as an energy blockade, preventing any energy growth in the central region $(x_*, L - x_*)$. We conclude that in such a regime the dissipative system will not tend to a steady state with uniform density, where particle density is 1 (0) if $\mathcal{R} < 1$ ($\mathcal{R} > 1$) (see Eq. (21)) , because of the persistence of the horizons which effectively decouple the system into three pieces: the left and right edges $[0, x_*]$ and $(L - x_*, L]$ where particles accumulate or deplete because of the exchange with the bath, and the middle piece $(x_*, L - x_*)$ where the dynamics is unaffected by the dissipative couplings to the baths. This decoupling is understood via the quasiparticle picture, where the micromotion of quasiparticles is concentrated around the horizons, and the system is effectively quenched with a deformed Hamiltonian with a

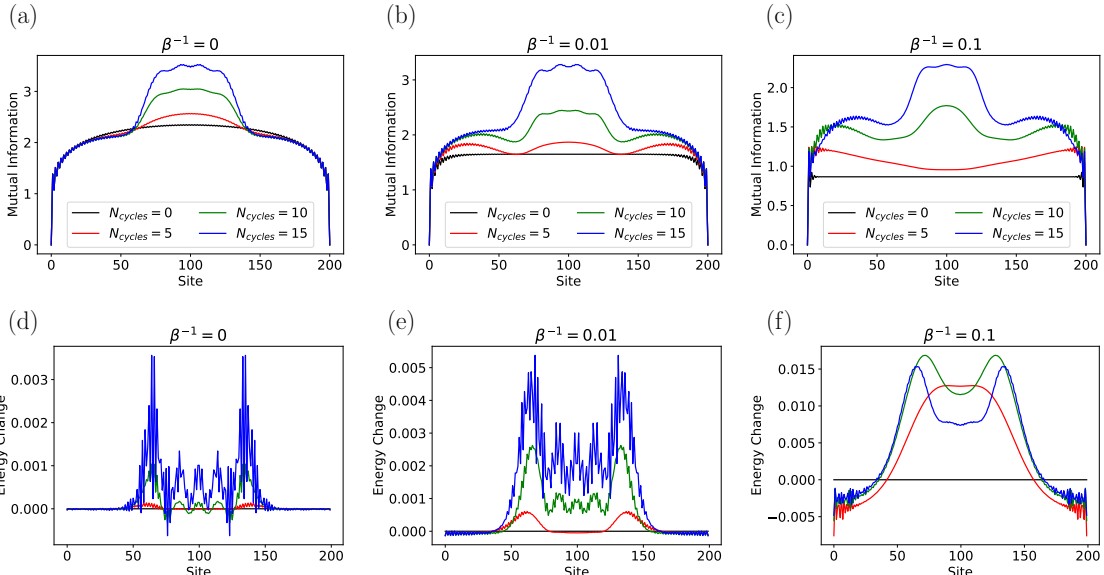

Figure 5: (a-c) Scaling of mutual information $I([0,x),(x,L])$ in the heating phase, $T_0/L = 0.95$, $T_1/L = 0.05$, $L = 200$ for different initial temperatures, $\beta^{-1} = 0, 0.01, 0.1$. A kink structure is observed at the position of the two horizons $x_* \approx 55$ and $L - x_*$, as predicted by CFT. (d-f) Energy density $E(x,t) - E(x,0)$ in the heating phase $T_0/L = 0.95$, $T_1/L = 0.05$, $L = 200$ for different initial temperatures, $\beta^{-1} = 0, 0.01, 0.1$. We observe at $x_*$ and $L - x_*$ two hotspots in energy density that are building up exponentially in time.

velocity profile $v_{\text{eff}}(x)$ such that $v_{\text{eff}}(x_*) = v_{\text{eff}}(L - x_*) = 0$, leading to such decoupling [39]. We note that this quenched dynamics is only a good approximation in the high-frequency limit $|T_0| + |T_1| \ll L$. Besides exponential energy accumulation, horizons also share mutual information $I(A,B)$ which grows linearly in time, as discussed in Sec. 2. This linear growth of mutual information is shown in Fig. 7(a). The extent of this linear regime increases with system size $L$, as clearly shown in Fig. 7(b). We see that the physics of entangled horizons persist for a substantial range of dissipation in large enough systems. However, we note that above a certain dissipation threshold of about $\gamma \sim 0.1$, we lose this linear regime of entanglement growth in the high-frequency regime.

Away from such a high-frequency limit, micro-motion is not negligible anymore [69] and quasiparticles can travel through the whole system in a single Floquet period, which makes the horizon picture only valid at stroboscopic times, and cannot decouple the system completely at all times. The energy density will ultimately, at long enough times, increase uniformly in the system instead of being confined to the horizons as we increase dissipation. Our results for the mutual information for the case $\Gamma_+^{L/R} = \Gamma_-^{L/R} =: \gamma$ are summarized in Fig. 8. We find that the mutual information is robust to small enough dissipation $\gamma < 0.005$, indicating that the entangled horizons survive for a range of times, even in cases where $|T_0| + |T_1| \sim L$. We stress that such a structure is not observed as clearly in the entanglement entropy, as it does not integrate out entanglement shared with the bath contrary to mutual information.

An important question concerns whether the transition from the heating to the non-heating phase is modified or smeared by dissipation. The heating phase is characterised by linearly growth of mutual information in time, as long as the subsystems $A$ and $B$ contain one of the two horizons each, while the non-heating phase has an oscillatory mutual information. Therefore we analyse the scaling of the half-system mutual information $I\left([0,\frac{L}{2}),(\frac{L}{2},L]\right)$ after 10 Floquet

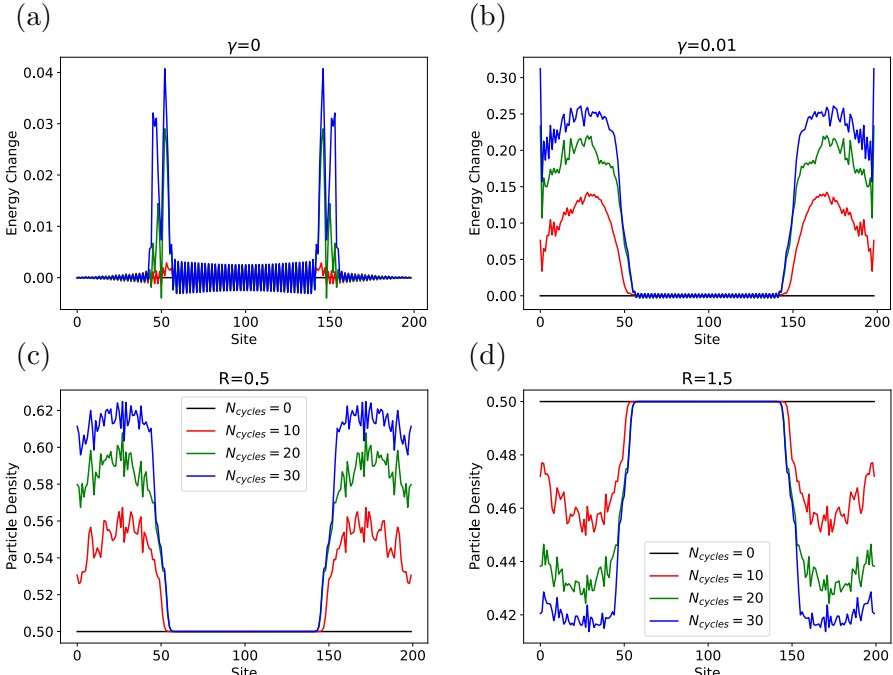

Figure 6: High-frequency regime for the driving parameters, $|T_0|/L = |T_1|/L = 0.05$. (a,b): Energy density time evolution $E(x,t) - E(x,0)$ in the case without dissipation, $\gamma = 0$, as well as the case with dissipation $\gamma = 0.01$. (c) Particle density evolution, for $\gamma = 0.01$, for different number of Floquet cycles, with $\mathcal{R} = 0.5$, leading to an overall particle gain. (d) Same as (c) but for $\mathcal{R} = 1.5$, leading to an overall particle loss.

cycles across the phase transition predicted by CFT across the $\frac{T_0}{L} = \frac{T_1}{L}$ line in Fig. 9(a). In the non-heating phase mutual information oscillates by varying the driving parameters, while it suddenly grows as a function of $\frac{T_0}{L}$ after $\frac{T_*}{L} \approx 0.415$, as predicted by CFT in the non-dissipative case, and then increases as a function of driving parameters in the heating phase. Although the critical exponent cannot be extracted clearly in the dissipative case, we still clearly observe a non-analyticity at $\frac{T_*}{L}$, signalling the persistence of the phase transition, for values of the dissipation rate $\gamma$ smaller than 0.005. After this threshold, dissipation dephases quasi-particles which makes it impossible to correctly observe entangled pairs of quasiparticles forming at the two horizons $x_*$ and $L - x_*$ at each Floquet cycles, leading to entanglement growth between left and right regions in the dissipationless case.

We now consider dephasing at the boundaries of the chain instead of letting particle exchange with an external bath. In contrast to the previous case, here the dissipation conserves particle number and the particle density will not show any signature of the horizons. We therefore focus on the energy density $E(x,t)$, computed from the correlation matrix, as well as the mutual information averaged over the Gaussian white noise realizations $\xi_i(t)$ in the presence of a boundary potential $V_i = \xi_i(t)n_i$, as explained in Sec. 3. We show the half-system mutual information after 10 drive cycles in Fig. 9 (b). While dephasing is expected to partially suppress the generation of entanglement since it drives the system into a trivial mixed state, we find that the half-chain entanglement entropy grows with increasing dephasing strength for several driving cycles. This shows that fluctuations induced by our dephasing protocol at the edges of the system dominate the dynamics on the time scales we are considering. Our results show that the entanglement dynamics introduced by the dephasing smoothens out the sharp transition from the heating-to-nonheating phase compared to the previous dissipation scheme. This growth in entanglement is also connected to the rapid growth of energy density

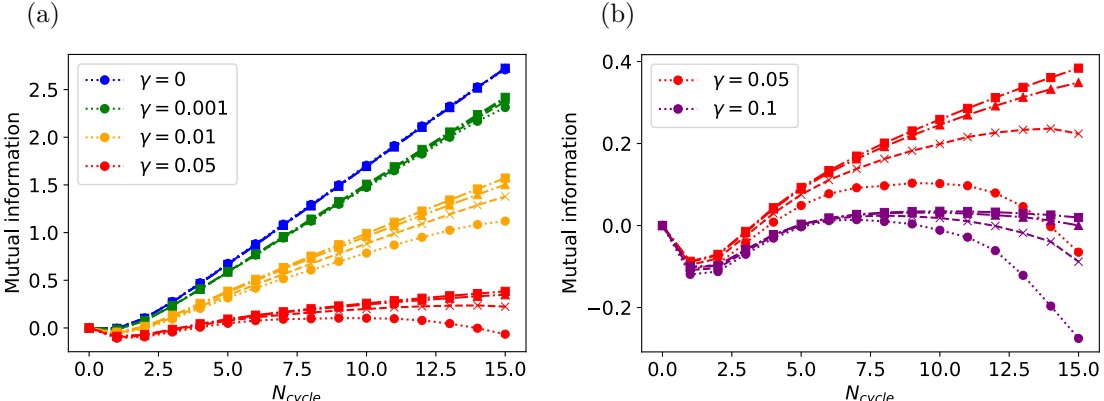

Figure 7: (a) Growth of half-system mutual information $I([0, L/2], [L/2, L])$ for $|T_0/L| = 0.05$, $T_1/L = 0.05$, with different dissipations, and system sizes $L = \{100, 200, 500, 800\}$ (circle, cross, triangle, square). (b) Same plot for different values of $\gamma$.

across the whole system, as shown in Fig. 10. This dynamics of the total energy stems from the injection of quasiparticle excitations due to the fluctuations at the edges and is generally found to be exponential in time [65, 70].

Although there is no sharp transition between the heating and non-heating phase, the energy density $E(x, t)$ in the heating phase still forms robust peak structures, similar to the heating phase in the closed system, see Fig. 10. The horizon structure persists for more than 10 driving cycles, despite dephasing strengths of the order of $\gamma = 0.001$. Furthermore, the horizons remain at the positions $x_*$ and $L - x_*$ predicted by the CFT, despite the sizable change in the total energy, as shown in Fig. 10(b).

We finally note that studying the interplay between dissipation and dephasing could lead to even richer physics. In systems without any Floquet driving, for example the XXZ chain, it has been shown that the interplay between edge dissipators and local dephasing terms can indeed result in varied regimes of heat and spin transport, for instance unidirectional heat flow, or heat flowing from both edge reservoirs towards the middle. Transitions from a ballistic regime to diffusive regime can be generated by such dissipators [71]. When coupled with periodic driving, it is highly likely that interesting regimes of behaviour, especially from a quantum thermodynamics perspective, might arise. It will be interesting to study in detail these effects.

## 6   Conclusion

We studied the robustness of the heating-to-non-heating phase transition of a particular class of driven integrable systems to the addition of temperature and dissipation. In particular we considered a free fermionic chain, whose low energy physics is described by a $c = 1$ CFT, periodically driven out-of-equilibrium using spatially deformed Hamiltonians. In the closed setting, starting from a pure state, the physics of this problem is well-described by an integrable Floquet CFT problem which is exactly solvable and predicts distinct heating and non-heating phases. We first analytically generalized this result to thermal initial states, showing that the Floquet CFT predictions extend to finite temperature up to an overall prefactor in the energy evolution, leaving the scaling of heating rate and periodicity unchanged. We then studied the case of a thermal initial state on the lattice, which still leads to distinct phases with an energy time evolution well-predicted by CFT for a wide range of temperatures, and shows the correct

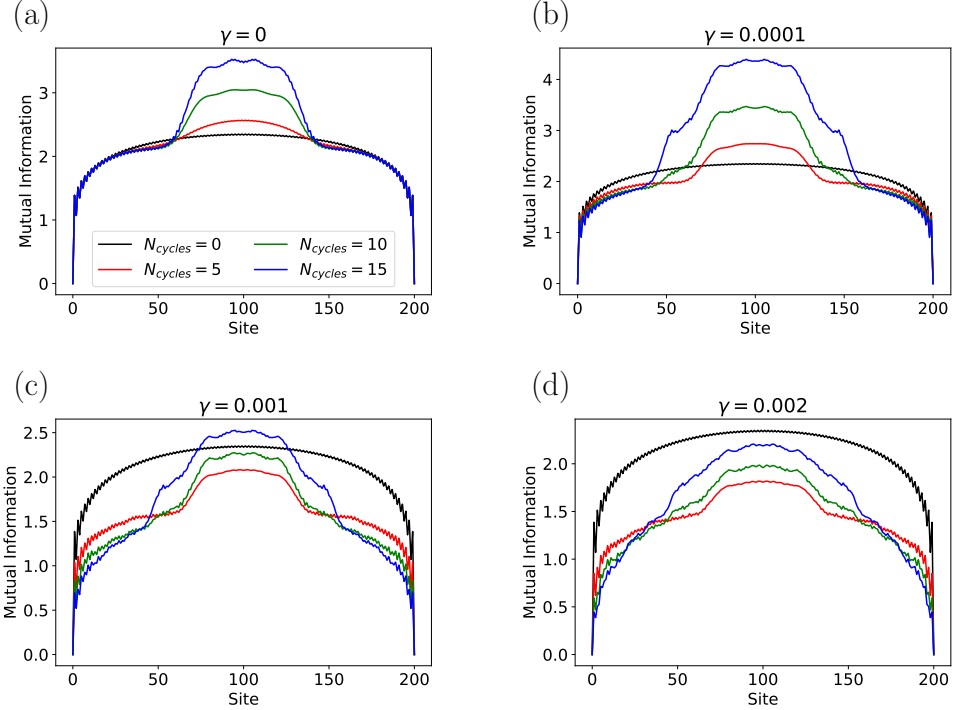

Figure 8: Scaling of mutual information in the heating phase $T_0/L = 0.9$, $T_1/L = 0.1$ and $L = 200$, for different values of dissipation $\gamma = \Gamma_+^L = \Gamma_+^R$, with $\Gamma_+^{L/R} = R\Gamma_-^{L/R}$ with $R = 0.1$, (a) no dissipation, $\gamma = 0$, (b) $\gamma = 0.0001$, (c) $\gamma = 0.001$, (d) $\gamma = 0.002$.

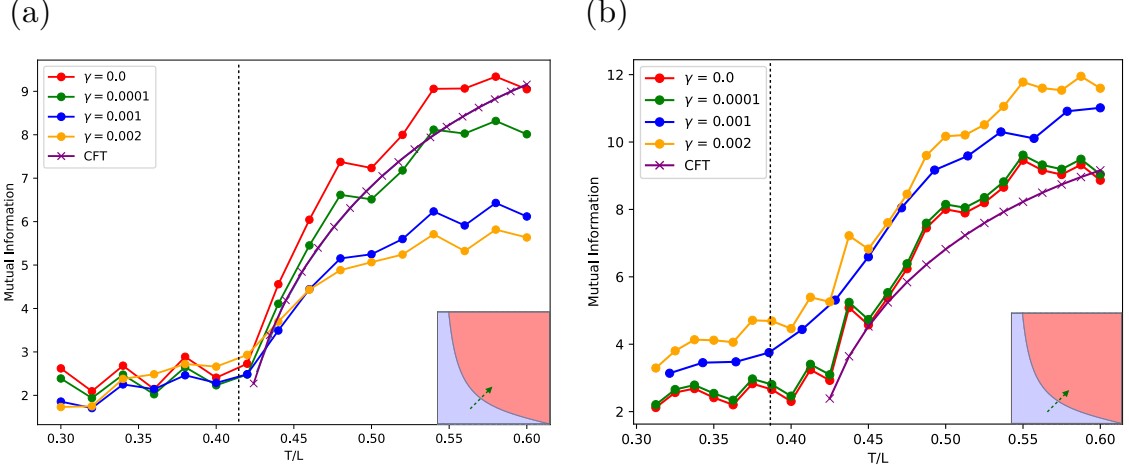

Figure 9: (a) Half-system mutual information $I\left([0, \frac{L}{2}), (\frac{L}{2}, L]\right)$ after 10 Floquet cycles as a function of $\frac{T_0}{L} = \frac{T_1}{L} =: \frac{T}{L}$ for different values of the dissipation $\gamma$, and $L = 100$. The phase transition in the CFT model occurs at $\frac{T_*}{L} = 0.415$. We also display the scaling predicted by the CFT for the mutual information in the dissipationless case. We observe a kink in the mutual information around $\frac{T_*}{L}$, signalling the persistence of the heating and non-heating phases in the entanglement structure of the system, even for non-zero dissipation. (b) Same figure but with dephasing instead of particle exchange with the bath.

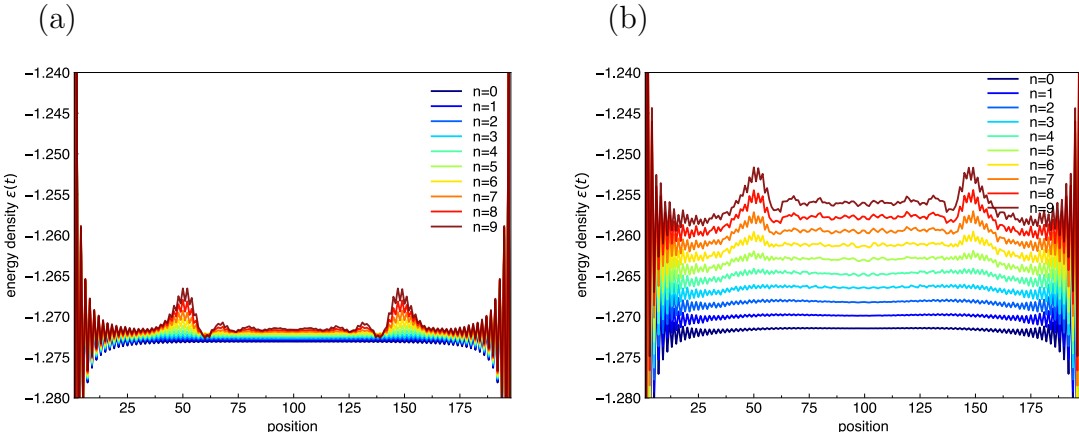

Figure 10: Time evolved energy density $E(x,t)$ in the heating phase with driving parameters $T_0/L = 0.95$, $T_1/L = 0.05$ for the dephasing protocol on the boundaries, with dephasing strengths (a) $\gamma = 0.0001$ and (b) $\gamma = 0.001$. The horizon structure of the heating phase persists in presence of dissipation.

critical exponent of $\frac{1}{2}$ of the order parameter at the phase transition. We found new effects arising on the lattice that are not predicted by CFT, such as temperature-dependent energy damping in the non-heating phase. We then considered two explicit open setting scenarios for the dissipation: either by exchange of particles with two external baths placed at the boundaries of the chain, or by adding on-site dephasing at the boundaries. While the first case does not conserve particle number, sharp signatures of the phase transition of the dissipationless case survive in the entanglement entropy structure of the heating phase, even beyond the high-frequency limit. On the other hand under the addition of on-site dephasing, that conserves particle number, the particular spatial structure of the energy density still survives ranges of boundary dephasing, giving a clear sign of the robustness of the emergent energy peaks that only depend on the driving parameters. However the physics of these two types of dissipative effects differs significantly: for dissipative particle loss, in the high-frequency regime of the heating phase, the horizons blockade both energy and particle flow through the system. In the low frequency regime, substantial micromotion smears out this blockade. Nonetheless, mutual information still remains sensitive to the creation of these horizons. In the dephasing case however, the horizons still act as energy hotspots, akin to the closed system. We summarize our findings in Fig. 11. Fig. 11(a) shows the impact of finite temperature as well as micromotion on the dynamics of entanglement, while Fig. 11(b) shows the impact of dissipation on the growth of entanglement, in both high and low-frequency driving regimes.

While we focused only on non-interacting lattice models in this work, we expect that our results should still survive (i) in interacting critical lattice models whose low-energy description is also a CFT (see [39] for a study of the driven XXZ model in the dissipationless setting) and (ii) to general spatial deformations of the energy density [45,46]. While the current work focused on one spatial dimension, extensions to higher dimensions are conceivable, as the sine-square deformation only involves the global conformal group that is present in CFTs of any dimension. In higher dimensions, we expect the driven dynamics to still exhibit heating and non-heating phases, however the heating pattern might not only consist of discrete points in space, leading to richer physics of "integrable heating". Lastly, from a field theory perspective, the persistence of the heating features in a dissipative setting which is inherently non-unitary, suggests that this physics might be present in a non-hermitian setting. This naturally paves the way to generalizations of inhomogeneous and interacting Floquet CFTs to non-unitary CFTs. Some recent efforts in this direction were initiated in [43]. Such an approach would be

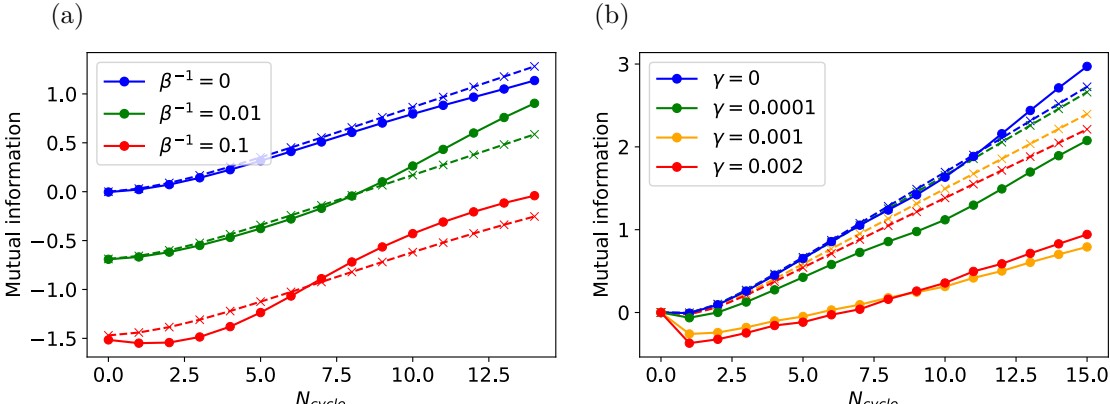

Figure 11: (a) Growth of half-system mutual information $I([0, L/2], [L/2, L])$ for $T_0/L = 0.95$, $T_1/L = 0.05$ (full lines), $T_0/L = -0.05$, $T_1/L = 0.05$ (dashed lines), and different initial temperatures. The equilibrium value of half-system mutual information in the ground state has been subtracted. (b) Growth of half-system mutual information $I([0, L/2], [L/2, L])$, for $T_0/L = 0.9$, $T_1/L = 0.1$ (full lines), $T_0/L = -0.1$, $T_1/L = 0.1$ (dashed lines), and different dissipations $\gamma$.

especially useful to explore the possibility of new classes of dissipative phase transitions corresponding to novel fixed point field theories in non-unitary CFTs stemming from a complex interplay between interactions, drive and dissipation.

## Acknowledgements

This project has received funding from the European Research Council (ERC) under the European Union's Horizon 2020 research and innovation program ERC- StG-Neupert-757867-PARATOP. AT is supported by the Swedish Research Council (VR) through grants number 2019-04736 and 2020-00214 and by the European Union's Horizon 2020 research and innovation program under the Marie Sklodowska-Curie Grant Agreement No. 701647.

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
