# Peer review of "Thermal and dissipative effects on the heating transition in a driven critical system"

_SciPost Physics, doi:SciPost Phys. 13, 104 (2022)_

## Round 1 · Referee Report · Anonymous (Referee 1) · 2022-6-29

Report

The authors consider a free fermion chain subject to a binary drive and Markovian dissipation at the boundaries. The correlation matrix for this model satisfies a closed equation of motion and is readily integrated numerically. The authors analyze the physical properties of this system, having in mind that in a particular parameter regime it is expected to be well described by a (previously studied) driven conformal field theory. The main conclusion of the work is that the heating-to-non-heating transition observed previously in the Floquet CFT setting appears to be robust against lattice effects, temperature and dissipation at the boundaries of the system. Studying the robustness of the Floquet-CFT predictions is definitely a good idea, and using a model amenable to analysis by means of free fermion techniques is a very sensible first step. The findings presented in this work definitely warrant publication, but it is not entirely clear to me that it meets the stringent acceptance criteria for SciPost Physics. From a methodological point of view the paper is in my view very straightforward, while on the other hand no unexpected new phenomena are observed. However, I am sure the authors would argue that this absence is in itself a key result of the work (and I would agree with this).

I have a couple of questions/comments:

  1. In Fig. 3 ten cycles are considered. An obvious question is how these pictures change as the number of cycles is increased.
  2. In Fig. 2 much lower temperatures are considered than in Fig.2. This is unfortunate in the sense that Fig. 3 (b) and (c) cannot be related to any curves in Fig. (2).
  3. The dissipative with particle loss/gain is special because the density matrix remains Gaussian (i.e. the full model is integrable). This ceases to be the case for the dephasing noise. A question I have is whether there are any interesting effects if both types of dissipative couplings are present, i.e. particle/gain loss and dephasing on top (making the model non-integrable).
  • validity: top
  • significance: good
  • originality: good
  • clarity: top
  • formatting: excellent
  • grammar: excellent

Author:  Bastien Lapierre  on 2022-09-01  [id 2783]

(in reply to Report 1 on 2022-06-29)

Dear Referee,

Please find attached to this comment the reply to your report.

Sincerely,
The authors

Attachment:

reply_referee1.pdf

---

## Round 1 · Referee Report · Anonymous (Referee 2) · 2022-7-6

Strengths

  • Nice and clear overview of previous results
  • Interesting topic

Weaknesses

  • Poor numerical analysis

Report

The authors consider periodically driven 1d critical models and study the effects on dynamics and, more specifically, on the heating transition when starting from thermal initial states and when coupling the system to some external bath. The underlying question is whether the Floquet-CFT phenomenology, valid for periodically driven systems starting from a pure zero-temperature state, is robust to such thermal and dissipative effects. To address such question, the authors use exact free fermionic calculations on the lattice. They use a bunch of semi-analytical techniques to write the evolution of the two-point correlation function, which enables to compute all different observables (given the free nature of the problem).

The paper is self-contained, providing a nice overview of the Floquet-CFT technique and the way to keep into account the dissipation. My main complaint, though, is about the presentation of new material. Specifically, while it is certainly of interest and I find the results non-trivial, the numerical analysis could definitely be improved. More specifically:

  • Fig. 5 and Fig. 7 show the mutual information as a function of the subsystem site (if I understand property). Since the claim the author make is about a linear growth in the stroboscopic time (within the central region), it would be useful to see the plot of mutual information vs N_{cycles}, to check that such linear growth
  • The authors comment about energy accumulation at x and (L-x) being visible in Fig. 6b. However, it does not seem to me that a clear pick is there, but rather a whole region extending from the edges to those points. Maybe they could show more values of gamma and/or comment more about that.
  • The authors claim the CFT prediction being robust to dissipation for values of gamma<0.005. This value is however very small. I wonder whether this means that this robustness only holds in a certain scaling limit. To this aim it would have seem natural to me to investigate this gamma-range as a function of the system/subsystem size.

A part that, I also have some revision suggestions:

  • In the introduction: “Exceptions to this paradigm include some integrable disordered systems [30] as well as many-body localized systems [31] which evade heating.” Maybe the authors meant “do not evade heating”? I think that at least Ref.30 is about some integrable disorder system is shown to heat up.
  • Capture Fig. 1: “We want to understand the robustness of this phenomenology to the introduction of dissipation”. I would stress that also thermal effects are of interest here.
  • Section 2: “Using the tools developed in Ref. [47,48]”. I think that the transformations in Eq. 2 were first given in this context in SciPost Phys. 2, 002 (2017) and SciPost Phys. 6, 051 (2019).
  • Section 4 (below eq. 27): h_0 -> H_0 ?
  • Section 4: About the extension of the Floquet-CFT results to finite temperature the authors claim “This complexity can be traced to the fact that for thermal states, the correlation functions cannot be fixed solely by conformal invariance, but require the knowledge of the full operator content of the theory at hand ”. I honestly do not understand this point as, to my knowledge, zero and finite temperature correlations are simply related by a conformal transformation. I am probably missing something, but I think the authors could try to explain this better.

Once these improvements are included, I’m happy to recommend the paper for publication in Scipost.

Requested changes

(see report)

  • validity: good
  • significance: good
  • originality: high
  • clarity: high
  • formatting: excellent
  • grammar: excellent

Author:  Bastien Lapierre  on 2022-09-01  [id 2784]

(in reply to Report 2 on 2022-07-06)

Dear Referee,

Please find attached to this comment the reply to your report.

Sincerely,
The authors

Attachment:

reply_referee2.pdf

---

## Round 2 · Referee Report · Anonymous (Referee 3) · 2022-9-1

Report

I'm happy with the author's reply. Therefore I recommend the paper for publication in its current form.

---

## Round 2 · Referee Report · Anonymous (Referee 4) · 2022-9-6

Report

The authors have made a number of changes to the manuscript, which have strengthened it. In particular they have addressed the issues I flagged with regards to Figs 1 and 3. In their reply to my first report they state their case for publication in SciPost Physics, and on balance I am minded to agree that the manuscript warrants publication there.

---

## Round 2 · Author Response

Dear Editor,

For ease of reading, as our reply contains several figures, we wrote it as a pdf file which can be accessed here:

https://drive.google.com/file/d/1k47qEKVWy_3M-25yroyJaUmJFpGKC9fA/view

Sincerely,
The authors

---

## Round 2 · List of Changes

1. The abstract and introduction were partially rewritten to make explicit mention of the new results on the CFT side at finite initial temperature.

2. Caption of Fig. 1 was changed according to the comment from Referee 2.

3. The Refs. [47,48] were added to the manuscript, as suggested by Referee 2.

4. Section 4 was fully rewritten. In particular, we added Sec. 4.1 "Thermal Floquet CFT" that deals with the analytic computation of the energy evolution starting from thermal initial state using CFT techniques.

5. Figure 2 was changed to make direct comparison between CFT and lattice results at finite temperature. Plots showing long time evolution of the energy in the non-heating phase were also added in Fig. 2(c,d).

6. In Sec. 4.2, the analysis of the total energy growth was rewritten to discuss CFT comparison as well as energy damping effects in the non-heating phase.

7. Figure 3 was changed according to the comment of Referee 1: the initial temperatures now displayed are $\beta=\{0,0.05,0.1\}$, in order to have explicit comparison between Fig. 3(a,b) and Fig. 2.

8. Figure 7 was added as suggested by Referee~2, together with a paragraph on page 16.

9. A paragraph was added on page 18 about adding both dissipation and dephasing, as suggested by Referee 1

10. The conclusion was partially rewritten to take into account the new changes in both Sec. 4 and 5. In particular, Figure 11 was added, as suggested by Referee 2.

---

## Editorial Decision

published